# Low-level *Plasmodium vivax* exposure, maternal antibodies, and anemia in early childhood: Population-based birth cohort study in Amazonian Brazil

**Anaclara Pincelli**[1], **Marly A. Cardoso**[2], **Maíra B. Malta**[2,3], **Igor C. Johansen**[1], **Rodrigo M. Corder**[1], **Vanessa C. Nicolete**[1], **Irene S. Soares**[4], **Marcia C. Castro**[5], **Marcelo U. Ferreira**[1]*, on behalf of the MINA-Brazil Study Working Group[¶]

1 Department of Parasitology, Institute of Biomedical Sciences, University of São Paulo, São Paulo, Brazil, 2 Department of Nutrition, School of Public Health, University of São Paulo, São Paulo, Brazil, 3 Graduate Program in Collective Health, Catholic University of Santos, Santos, Brazil, 4 Department of Clinical and Toxicological Analyses, School of Pharmaceutical Sciences, University of São Paulo, São Paulo, Brazil, 5 Department of Global Health and Population, Harvard T.H. Chan School of Public Health, Boston, Massachusetts, United States of America

¶ A full list of members of the MINA-Brazil Study Working Group is provided in the Acknowledgments section.
* muferrei@usp.br

**Data Availability Statement:** All relevant data are within the paper and the original, anonymized database has been deposited in the WorldWide

## Abstract

### Background

Malaria causes significant morbidity and mortality in children under 5 years of age in sub-Saharan Africa and the Asia-Pacific region. Neonates and young infants remain relatively protected from clinical disease and the transplacental transfer of maternal antibodies is hypothesized as one of the protective factors. The adverse health effects of *Plasmodium vivax* malaria in early childhood–traditionally viewed as a benign infection–remain largely neglected in relatively low-endemicity settings across the Amazon.

### Methodology/Principal findings

Overall, 1,539 children participating in a birth cohort study in the main transmission hotspot of Amazonian Brazil had a questionnaire administered, and blood sampled at the two-year follow-up visit. Only 7.1% of them experienced malaria confirmed by microscopy during their first 2 years of life– 89.1% of the infections were caused by *P. vivax*. Young infants appear to be little exposed to, or largely protected from infection, but children >12 months of age become as vulnerable to vivax malaria as their mothers. Few (1.4%) children experienced ≥4 infections during the 2-year follow-up, accounting for 43.4% of the overall malaria burden among study participants. Antenatal malaria diagnosed by microscopy during pregnancy or by PCR at delivery emerged as a significant correlate of subsequent risk of *P. vivax* infection in the offspring (incidence rate ratio, 2.58; *P* = 0.002), after adjusting for local transmission intensity. Anti-*P. vivax* antibodies measured at delivery do not protect mothers from subsequent malaria; whether maternal antibodies transferred to the fetus reduce early malaria risk in children remains undetermined. Finally, recent and repeated vivax malaria episodes in

Antimalarial Resistance Network (WWARN) Data Platform, hosted by the University of Oxford, as part of the Impact of Malaria in Pregnancy on Infants Study Group. The Terms of Use of these data can be found at www.wwarn.org. Investigators must direct their data access requests through the WWARN Data Access Committee (DAC), the membership and details of which can be seen at www.wwarn.org/DAC.

**Funding:** Supported by the Conselho Nacional de Desenvolvimento Científico e Tecnológico (CNPq; www.cnpq.br), Brazil (grant 407255/2013-3 to M. A.C.); the Maria Cecília Souto Vidigal Foundation (https://www.fmcsv.org.br/en-US/; grant to M.C. C.), Brazil; and the Fundação de Amparo à Pesquisa do Estado de São Paulo (FAPESP; www. fapesp.br), Brazil (grant 2016/00270-6 to M.A.C.). A.P. (2018/18557-5), M.B.M. (2017/05019-2), and V.C.N. (2020/07020-0) are or were supported by FAPESP scholarships; M.A.C., R.M.C., M.C.C., I.S. S., and M.U.F. receive or received CNPq scholarships. The funders had no role in study design, data collection and interpretation, or the decision to submit the work for publication.

**Competing interests:** The authors have declared that no competing interests exist.

early childhood are associated with increased risk of anemia at the age of 2 years in this relatively low-endemicity setting.

## Conclusions/Significance

Antenatal infection increases the risk of vivax malaria in the offspring and repeated childhood *P. vivax* infections are associated with anemia at the age of 2 years.

## Author summary

*Plasmodium vivax* malaria causes frequent hospital admissions of infants and toddlers in areas of intense transmission in the Asia-Pacific region, often due to severe anemia, but its epidemiology and burden have been understudied in children from other endemic settings. Here we characterize the cumulative impact of *P. vivax* infections in infants and toddlers exposed to relatively low levels of malaria transmission in the Brazilian Amazon. We have previously shown that vivax malaria in pregnancy is associated with increased risk of maternal anemia and impaired fetal growth in this population. Now we show that the adverse effects of malaria extend to early childhood. Children born to mothers who had one or more infections during pregnancy are at an elevated risk of *P. vivax* malaria in their early life, although the transfer of maternal antibodies to the fetus may provide some short-term protection. Children who are repeatedly infected with *P. vivax* since birth are more likely to be anemic at the age of 2 years. These findings further challenge the traditional view of vivax malaria as a relatively benign infection in pregnancy and early childhood in the Amazon.

## Introduction

Malaria transmission has decreased substantially in Latin America and the Caribbean over the past two decades, but 120 million people remain exposed to some risk of infection across this region [1]. *Plasmodium vivax* accounts for 72% of the 889,000 malaria infections reported yearly in the Americas [2] and appears to be more resilient than *P. falciparum* to current control and elimination strategies worldwide mostly due to relapses and early circulation of mosquito-infective blood stages, the gametocytes [3].

*P. falciparum* infection is a leading cause of morbidity and mortality in children exposed to intense transmission in sub-Saharan Africa [4], but neonates and young infants are relatively protected from clinical disease. The poorly understood protective mechanisms appear to comprise maternal antibodies transferred to the fetus in the last trimester of pregnancy, the relative inability of fetal hemoglobin to support intraerythrocytic parasite development, the deficiency of *p*-aminobenzoic acid in exclusively breastfed infants, and the reduced exposure of young children to mosquito vectors [5]. Infants >6 months of age become gradually vulnerable to repeated infections and children until the age of five years are heavily affected by severe malaria. Partial immunity develops over years and protects most adolescents and adults from clinical disease [6,7].

*P. vivax* is the second most prevalent human malaria parasite worldwide [2]. It is mostly absent from Central and West Africa, but causes significant morbidity in children elsewhere in the tropics [8–10]. More malaria-related hospital admissions in infants and toddlers are due to *P. vivax* than *P. falciparum* in the Asia-Pacific region, where both species are co-transmitted

[11,12]. Severe anemia is particularly common in young children with malaria, regardless of the infecting parasite species [12,13].

The adverse health effects of vivax malaria in the course of the first 1000 days–from conception to child's second birthday–remain understudied in the Americas [14]. A critical knowledge gap refers to the association between maternal malaria in pregnancy and risk of subsequent *P. vivax* infection in the offspring. No population-based data are available for Brazil, a country that contributes 20% of malaria cases in Latin America and the Caribbean region [2]. Here we examine the burden of early life malaria in Juruá Valley, an Amazonian hotspot in northwestern Brazil that contributes nearly 18% of the country's malaria cases [15]. We describe the incidence of *P. vivax* infection during the first two years of life in a prospective birth cohort study and its association with maternal antibodies and anemia at the age of two years.

## Materials and methods

### Ethics statement

All mothers or their parents or guardians, if mothers were <18 years old, provided written informed consent. The study protocol was approved by the institutional review board of the School of Public Health, University of São Paulo (# 872.613, 2014).

### Study design and population

The Maternal and Child Health and Nutrition in Acre, Brazil (MINA-Brazil) study is a prospective, population-based birth cohort set-up to examine the impact of a wide range of early exposures on child growth and development in the Amazon [16]. Mother-baby pairs were enrolled at pregnancy in public antenatal clinics, or at birth in the Women and Children's Hospital of Juruá Valley–the only maternity hospital of Cruzeiro do Sul, where 96% of all local deliveries take place [17]. The study site is the most populous municipality of the Upper Juruá Valley, Acre State, next to the Brazil-Peru border (Fig 1). With 89,072 inhabitants estimated in 2020 by the Brazilian Institute of Geography and Statistics (IBGE) (https://www.ibge.gov.br/cidades-e-estados/ac/cruzeiro-do-sul.html), Cruzeiro do Sul has approximately 72% of its population classified as urban. The area has a typical equatorial humid climate and receives most rainfall between November and April. Malaria transmission occurs year-round and the annual malaria incidence (API; number of laboratory-confirmed clinical malaria cases per 1,000 people per year) was estimated at 231.9 cases per 1,000 inhabitants in 2016, the fourth highest among municipalities in Brazil [18]. The infant mortality rate in Cruzeiro do Sul has been estimated in 2017 at 11.3 deaths among children under one year of age per 1,000 live births (https://www.ibge.gov.br/cidades-e-estados/ac/cruzeiro-do-sul.html), compared to the country average of 12.8 deaths per 1,000 live births (https://www.ibge.gov.br/estatisticas/sociais/populacao/9126-tabuas-completas-de-mortalidade.html?edicao=23111&t=sobre).

### Baseline and follow-up assessments

Fig 2 shows the study flow chart. At delivery, mothers were interviewed to obtain demographic, lifestyle, and morbidity information. Data on selected household assets were combined to derive a wealth index [19]–a proxy of socioeconomic status. Information on the number of antenatal care visits attended, gestational age at delivery, type of delivery (vaginal or cesarean), and child's sex was retrieved from hospital records [16]. The diagnosis of malaria in pregnancy was defined by combining antenatal diagnosis by microscopy of Giemsa-stained capillary blood thick smears prepared during sick visits to public clinics and species-specific

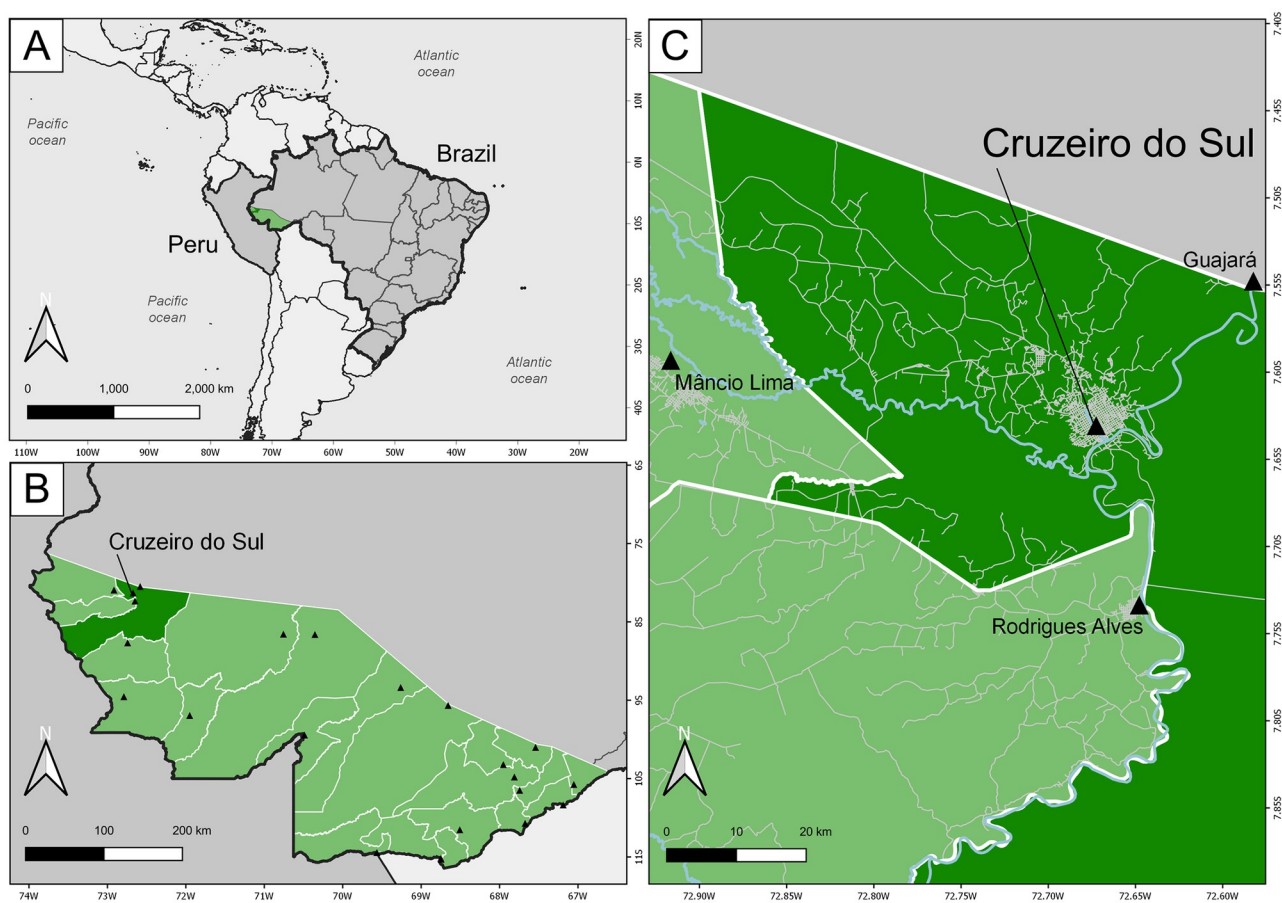

**Fig 1. Study site.** The map shows Brazil and Peru in South America (panel A) and the location of the municipality of Cruzeiro do Sul (dark green) in the Upper Juruá Valley region, Acre State (light green), northwestern Brazil (panel B). The urban area of Cruzeiro do Sul is shown in greater detail in panel C. Other cities and towns in the region (Mâncio Lima, Guajará, and Rodrigues Alves) are also indicated by triangles. Roads and streets are represented in light gray. Rivers are represented in blue. Figure created with QGIS software version 3.14, an open source Geographic Information System (GIS) licensed under the GNU General Public License (https://bit.ly/2BSPB2F). Publicly available shape files were obtained from the Brazilian Institute of Geography and Statistics (IBGE) website (https://bit.ly/34gMq0S). Roads, streets, and rivers were obtained from the Open Street Map Foundation website (https://bit.ly/3pzh4xp). All utilized geographical data are under the Creative Commons Attribution License (CC BY 4.0).

real-time PCR [20] carried out on venous blood samples collected at delivery [21]. Once infected with *P. vivax*, 23% of the pregnant women in this setting are estimated to have one or more vivax malaria recurrences over the next 12 weeks. The vast majority (86%) of these early *P. vivax* recurrences are likely relapses [22]. Soon after delivery, neonates had their birth weight measured by registered nurses to the nearest 0.005 kg, using a Toledo Junior portable scale (Toledo, São Bernardo do Campo, Brazil) with 15 kg capacity. Study participants were invited to attend a follow-up assessment at the age of 2 years, except for those living in remote rural communities (n = 302), typically with no electricity and cell phone coverage. These were considered ineligible because they could not be reached by telephone or social media. Here we analyze information collected during the follow-up visit at the age of 2 years. Participants in the assessment and those who were lost to follow-up had similar perinatal health profiles, but the proportion of participants from poorest families declined from 24.9% at birth to 19.3% at 2 years, mostly due to the exclusion of children from the remote rural sites [16]. Nurses and research assistants used structured questionnaires to update sociodemographic and morbidity information and ask about feeding practices. They collected 5-ml venous blood samples for a

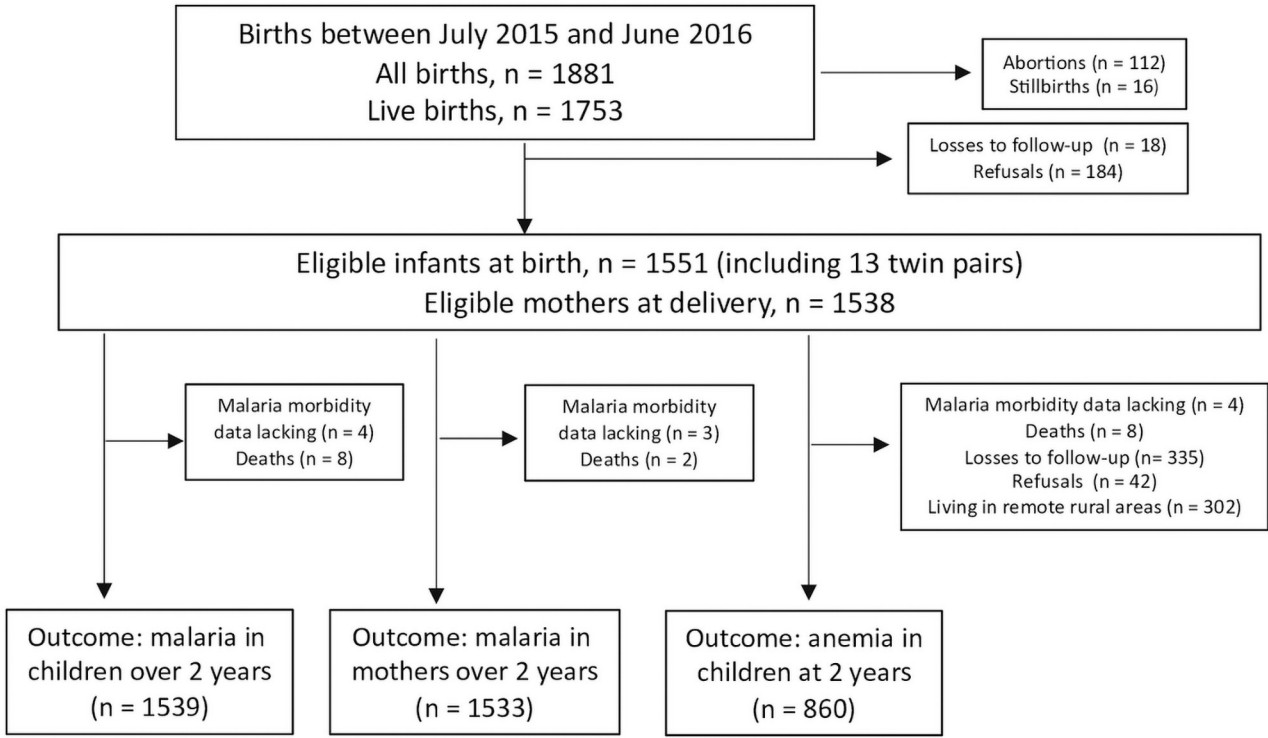

**Fig 2. Study flowchart.** Between July 2015 and June 2016, pregnant women attending antenatal clinics or admitted for delivery to the maternity ward of the Women and Children's Hospital of Juruá Valley in Cruzeiro do Sul, Brazil, were invited to participate. Reasons for exclusion and the final number of subjects analyzed for each study outcome are indicated.

range of laboratory measurements that included hemoglobin concentrations determined on an ABX Micro 60 automated cell counter (Horiba, Montpellier, France). Anemia was defined by a hemoglobin concentration <110 g/L (https://www.who.int/vmnis/indicators/haemoglobin.pdf).

## Malaria surveillance

We computed laboratory-confirmed clinical malaria episodes experienced by children from birth until the age of 2 years and by their mothers over the same period. Strategies used to obtain malaria morbidity data have been described elsewhere [22]. Briefly, we retrieved case notifications that matched children's name, sex, and age, in addition to their mothers' name and age, that were entered in the electronic malaria notification database of the Ministry of Health of Brazil (http://200.214.130.44/sivep_malaria/) between July 2015, when the first participants were born, and June 2018, when all study participants had already completed 2 years. The same strategy was used to retrieve clinical malaria cases in mothers up to 2 years after delivery. Access to this database has been granted by the Ministry of Health of Brazil to study the burden of malaria in pregnancy and early infancy in our study site [21]. Because malaria is a notifiable disease and diagnostic testing and antimalarial medications are not available in the private sector, the electronic database comprises virtually all laboratory-confirmed malaria episodes in Brazil [23]. At least 100 fields of Giemsa-stained thick blood smears are routinely examined for malaria parasites under 1000× magnification before a slide is declared negative [24]. *Plasmodium vivax* infections were treated with 25 mg/kg of body weight of chloroquine administered orally over three consecutive days and 3.5 mg/kg of primaquine administered

over 7 days. Primaquine was omitted in children over 6 months of age, as recommended by the Ministry of Health of Brazil [25].

Cruzeiro do Sul has been divided, for operational malaria control purposes, into smaller geographic units, or "localities", with shared epidemiological and ecological characteristics [26]. The central points of the localities (typically a health post or school) were georeferenced using hand-held GPS receivers, all dwellings were identified and given a unique identifier, and all residents were enumerated during periodic census surveys carried out by the local malaria control program staff. We retrieved from the electronic malaria notification database the following locality-related information: (i) GPS coordinates, (ii) total population size, and (iii) number of locally acquired, laboratory-confirmed malaria episodes in the locality between 2016 and 2018. This information was used to calculate the average API per locality between 2016 and 2018, a proxy of malaria transmission intensity in the area of residence of study participants.

## Recombinant antigens and antibody measurement in maternal and cord blood plasma

We used three well-characterized blood-stage parasite antigens expressed as recombinant proteins to capture anti-*P. vivax* maternal IgG antibodies: (i) residues 43–487 of the *P. vivax* apical membrane protein (PvAMA1) ectodomain, Brazilian strain, expressed in *Pichia pastoris* [27]; (ii) residues 194–521 of the *P. vivax* Duffy-binding protein (PvDBP) erythrocyte-binding domain, Salvador-I strain, expressed in *Escherichia coli* [28] and kindly provided by Christopher L. King (Case Western Reserve University, Cleveland, OH); and (iii) the C-terminal, 19-kDa region of *P. vivax* merozoite surface protein (MSP) 1 (PvMSP1$_{19}$), Belém strain, expressed in *E. coli* [29]. Methods for protein expression, purification, and refolding, if required, have been described elsewhere [27–29]. We used enzyme immunoassays (ELISAs) to measure antigen-specific IgG antibodies in 1,095 mothers (71.4% of the total) who provided plasma samples at delivery. Briefly, high-binding 96-well Costar microplates (Corning, Corning, NY) were coated with 50 μL of solid-phase antigen solution in phosphate-buffered saline for 18 h at 4°C. Antigen concentrations optimized for the maximal signal to background ratio were as follows: PvAMA1, 1 μg/mL; PvDBP, 0.025 μg/mL; and PvMSP1$_{19}$, 0.025 μg/mL. Plasma samples (50 μL/well) were tested at a 1:800 dilution for PvAMA1 and PvMSP1$_{19}$, or 1:200 for PvDBP. After a 1-h incubation at 37°C, antibody binding to solid-phase antigen was detected with peroxidase-conjugated goat anti-human IgG (SouthernBiotech, Birmingham, AL) at a 1:8,000 dilution. After the use of tetramethylbenzidine and hydrogen peroxide at acidic pH as a substrate, absorbance values were measured at 450 nm. Reactivity indices (RIs) were calculated as the ratio between the absorbance values of each test sample and a cutoff value for each antigen, corresponding to the average absorbance for plasmas from 10 malaria-naïve donors tested on each microplate plus 3 standard deviations. Positive samples had RIs greater than 1. To test whether maternal antibodies were efficiently transferred to infants, we compared antibody reactivity indices in a subset of 101 randomly selected paired maternal and cord-blood plasma samples from study participants. Importantly, maternal and cord blood antibody levels were strongly correlated to each other over the entire range of values (S1 Fig), consistent with no saturation of the placental FcRn receptor [30] at high maternal IgG levels.

## Statistical analysis

Data collected during the baseline and follow-up assessments were entered into tablets programmed with CSPro software (https://www.census.gov/programs-surveys/international-programs.html) and transferred to STATA 15.1 (StataCorp, College Station, TX) for statistical

analysis. Standard descriptive statistics were used to summarize the main study outcomes. Proportions were compared using Mantel-Haenszel $\chi^2$ tests for linear trend and correlations were evaluated using the nonparametric Spearman correlation test. Statistical significance was defined at the 5% level. We first computed the monthly incidence of laboratory-confirmed clinical malaria experienced by children (n = 1,539) until the age of 2 years and by their mothers (n = 1,533) up to 2 years after delivery, with their respective 95% confidence intervals (CIs).

We used negative binomial regression models to identify correlates of malaria risk among children. This analysis comprised all children assessed at delivery, except for those with unavailable API estimates for their areas of residence (n = 45). This gives a total of 1,494 children for whom malaria morbidity data from birth to their second birthday were retrieved from the case notification database, or 2,988 person-years of follow-up. Separate regression models were built for two continuous outcomes: (i) total number of laboratory-confirmed clinical malaria episodes among children, irrespective of the infecting parasite species, during their first two years of life and (ii) number of laboratory-confirmed vivax clinical malaria episodes among children during the same period, excluding mixed-species (*P. vivax* plus *P. falciparum*) infections. We excluded mixed-species infections because, in these cases, malaria-related morbidity could be associated with either species or to the interaction between them. Models included the sociodemographic, environmental, gestational, and neonatal covariates listed in S2 Fig and described in detail in S1 Text. We note that: (i) one of the socioeconomic covariates is whether the mother is beneficiary of the *Bolsa Família* program [31] because this Federal conditional cash transfer strategy targets selectively low-income families although not necessarily all poor families participate in it; (ii) gestational night blindness [32] was used as a proxy of vitamin A deficiency; (iii) gestational weight gain was categorized as insufficient, adequate, or excessive as described in [33], and (iv) gestational age at birth was estimated as described in [34]. Variables associated with the outcome at a significance level <20% in unadjusted analysis were further assessed in negative binomial models. We next used a hierarchical approach based on conceptual frameworks with distal, intermediate and proximal levels of disease determination (S2 Fig) [35], to select covariates that were retained in the final adjusted models. At each level of determination, covariates were retained in downstream analyses if they were associated with the outcome at a significance level of <10% or if their inclusion in the model changed the risk measures by ≥10%. Participants with missing values in categorical covariates were maintained in the model by creating a new missing-value category. We provide estimates of incidence rate ratio along with 95% CIs to quantify the influence of a given predictor on the outcome, while controlling for all other covariates retained in the final model, either in the same or more distal hierarchical level.

We used survival analysis to test whether levels of specific maternal antibodies measured at delivery predicted vivax malaria risk in mothers and in their children over the next months. We used Cox proportional hazards models to compare hazard ratios (HRs) for the time to the first vivax malaria episode after delivery in mothers and the first malaria episode experienced by their children (n = 1,095) across quintiles of maternal antibody levels, while adjusting for potential confounders. The first quintile (reference) comprises 20% of the study mothers with the lowest RIs. We used the hierarchical framework of disease determination described in S2 Fig to select for covariates to be retained in the final multiple models. The proportional-hazards assumption of Cox models was tested on the basis of Schoenfeld residuals after fitting the final model using the phtest STATA command. Because the proportional-hazards assumption was not fulfilled for mothers over the entire 24-month period, we carried out separate analyses for the periods (i) between 0 and 12 months after delivery (n = 1095) and (ii) the period after 12 years after delivery (n = 981, excluding mothers who had malaria up to 12 months after delivery). Mothers who experienced malaria were excluded from the second analysis because

their specific antibody status is more likely to have changed following a recent reexposure to the parasite.

We next used multiple logistic regression analysis to test whether previous malaria episodes in children were associated with an increased risk of anemia at the age of 2 years (dichotomic outcome variable). This analysis comprised 860 children who had hemoglobin measurements at the age of approximately 2 years (average, 725.9 days; standard deviation, 44.5 days). Covariates included in the multiple logistic regression models and the hierarchical model for variable selection are shown in S3 Fig; all models controlled for age to account for individual age variation, since children were not exactly 2 years old. To explore the impact of the frequency and timing of malaria episodes, as well as the infecting malaria parasite species, we considered four malaria exposure categories among children in separate models: (i) number of laboratory-confirmed malaria episodes during the first two years of life, caused by any species (0, 1, or 2+), (ii) number of laboratory-confirmed vivax malaria episodes during the first two years of life (0, 1, or 2+), (iii) one or more malaria episodes, any species, ≤12 months before the follow-up visit (no vs. yes), and (iv) one or more vivax malaria episodes ≤12 months before the follow-up visit (no vs. yes). Estimates of odds ratios along with their 95% CIs were interpreted to reflect the magnitude of the observed association between malaria exposure and anemia while controlling for all other covariates.

Quantile regression was further used to explore the association between prior malaria and hemoglobin levels in children at the 25th, 50th, and 75th hemoglobin quantiles (corresponding to 114 g/L, 121 g/L, and 127 g/L, respectively). To this end, we used the *sqreg* STATA command to estimate β coefficients and their bootstrap 95% confidence intervals while adjusting for the covariates shown in S3 Fig. Quantile regression provides a way of estimating the association between exposures and continuous health outcomes that are not normally distributed [36]; this is the case of hemoglobin concentrations in our study children (Shapiro-Wilk test for normality, $P < 0.001$).

## Results

### Malaria incidence in early childhood and associated risk factors

Overall, 265 malaria episodes were diagnosed in 110 (7.1%) of the 1,539 study children who were followed up from birth through the age of 2 years. *P. vivax* accounted for 236 (89.1%) of all malaria episodes. Only uncomplicated clinical disease was observed and treated on an ambulatory basis; no child was hospitalized due to malaria. Mothers suffered a total of 497 malaria episodes during the same period, 81.3% of them due to *P. vivax*; 257 (16.8%) of the 1,533 mothers had at least one malaria episode during the follow-up.

During their first 12 months of life, children experienced much fewer vivax malaria episodes than their mothers, with little or no overlap between the 95% CIs of incidence estimates in children and mothers. Thereafter, children become as vulnerable to vivax malaria as their mothers (Fig 3A). Episodes of falciparum or mixed-species malaria remained similarly infrequent in mothers and children over the entire follow-up period (Fig 3B).

The overdispersed frequency distribution of malaria episodes per child, with the variance (0.612) largely exceeding the mean (0.172 episode/child), is properly described by a negative binomial function (S4 Fig). There was a range of 0 to 9 malaria episodes per child, with 0–8 vivax and 0–2 falciparum or mixed-species infections per child. The vast majority of children (92.8%) remained free of malaria until their second birthday, but 21 (1.4%) of them had ≥4 episodes and together accounted for 43.4% (n = 115) of all malaria episodes.

Malaria incidence was heterogeneously distributed among localities (Fig 4). Not surprisingly, study children living in urban or periurban localities typically experienced fewer or no

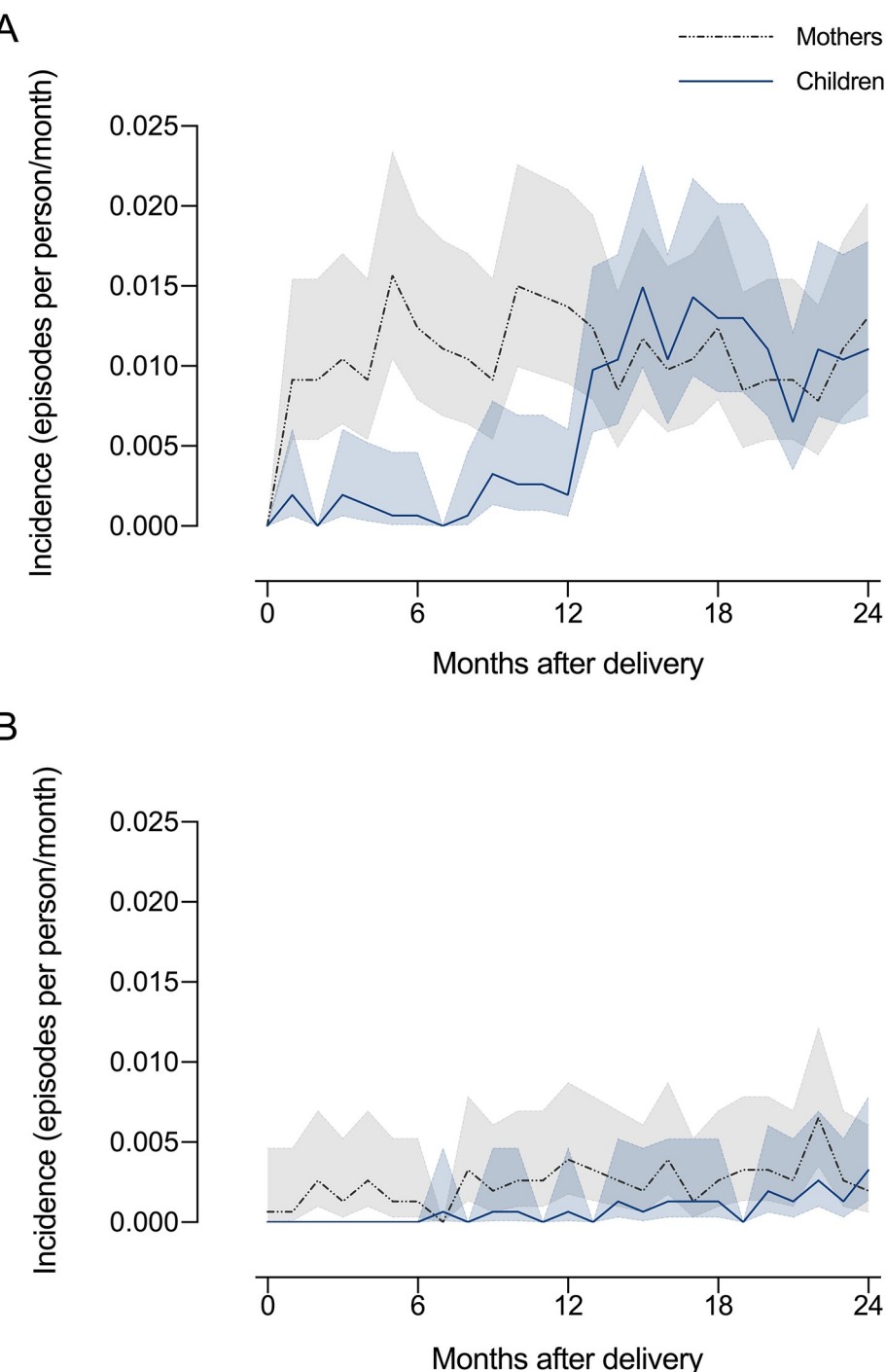

**Fig 3. Monthly malaria incidence in mothers and their children.** Panel A shows the incidence of laboratory-diagnosed *P. vivax* malaria and their respective 95% confidence intervals among mothers (n = 1,533) and their children (n = 1,539) participating in the MINA-Brazil study. Panel B shows the incidence of laboratory-diagnosed *P. falciparum* or mixed-species malaria in mothers and children. Each study participant contributed 2 person-years (24 person-months) of follow-up.

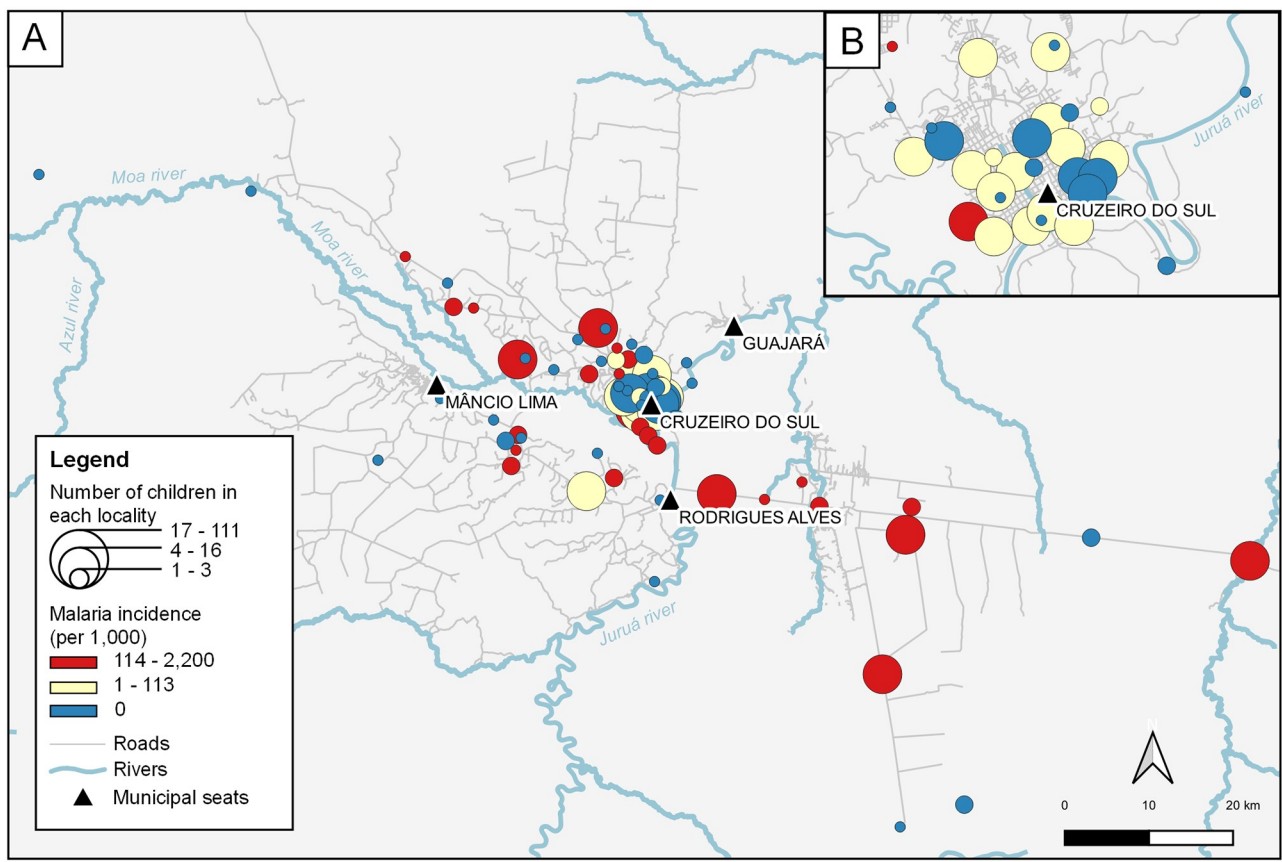

**Fig 4. Spatial distribution of malaria incidence among MINA-Brazil study children.** The map shows the upper Juruá Valley and the localities of residence of study children (circles) within and surrounding the municipality of Cruzeiro do Sul (A). The inset (B) shows the localities within and next to the urban area. Circle sizes are directly proportional to the number of study participants in each locality and their color (from blue to red) indicates the overall malaria incidence among study participants living in each locality (highest incidence in red). Roads and streets are represented in light gray. Rivers are represented in blue. Figure created with QGIS software version 3.14, an open source Geographic Information System (GIS) licensed under the GNU General Public License (https://bit.ly/2BSPB2F). Publicly available shape files were obtained from the Brazilian Institute of Geography and Statistics (IBGE) website (https://bit.ly/34gMq0S). Roads, streets, and rivers were obtained from the Open Street Map Foundation website (https://bit.ly/3pzh4xp). All utilized geographical data are under the Creative Commons Attribution License (CC BY 4.0).

malaria episodes, compared to their counterparts living in more remote rural localities surrounding the city of Cruzeiro do Sul.

Multiple negative binomial regression analysis identified the following significant predictors of malaria risk (any species) and vivax malaria risk in children (Table 1): (i) poverty (least poor children are at decreased risk compared with the poorest ones), (ii) API in the locality of residence, (iii) malaria in pregnancy, and (iv) maternal gravidity (children of secundigravidae are at decreased risk compared to primigravidae). Malaria in pregnancy, diagnosed in 178 (11.9%) mothers [21], was significantly associated with more than twice the risk of subsequent malaria even after controlling for transmission intensity in the children's area of residence. To further explore this association, we reran the negative binomial models with the key adjustment covariate API stratified into quintiles (S1 Text). These alternative models yielded very similar results (S1 Table).

## Maternal antibodies and risk of vivax malaria

We detected IgG antibodies to PvAMA1, PvDBP, and PvMSP1$_{19}$ in 346 (31.8%; 95% CI, 28.8–34.4%), 348 (31.7%; 95% CI, 29.0–34.6%), and 489 (44.4%; 95% CI, 41.7–47.7%) out of 1,095

**Table 1. Multiple negative binomial regression analysis of correlates of malaria (all species) and vivax malaria from birth to two years of age in children from the MINA-Brazil birth cohort study (n = 1494).**

| Covariate | n[a] | Outcome: all malaria species (n = 251) | | | Outcome: vivax malaria (n = 224) | | |
|---|---|---|---|---|---|---|---|
| | | IRR[b] | (95% CI)[c] | P | IRR[b] | (95% CI)[c] | P |
| **Wealth index quartile** | | | | | | | |
| 1 (poorest) | 341 | Reference | | | Reference | | |
| 2 | 358 | 0.635 | (0.339 1.189) | 0.156 | 0.641 | (0.336 1.224) | 0.178 |
| 3 | 363 | 0.597 | (0.314 1.135) | 0.115 | 0.635 | (0.328 1.230) | 0.178 |
| 4 (least poor) | 364 | 0.162 | (0.058 0.448) | <0.001 | 0.188 | (0.067 0.524) | 0.001 |
| **Beneficiary of *Bolsa Família*[d]** | | | | | | | |
| No | 821 | Reference | | | Reference | | |
| Yes | 605 | 1.956 | (1.137 3.364) | 0.015 | 1.977 | (1.133 3.449) | 0.016 |
| **API[e]** | 1494 | 1.003 | (1.002 1.004) | <0.001 | 1.003 | (1.002 1.004) | <0.001 |
| **Malaria in pregnancy** | | | | | | | |
| No | 1316 | Reference | | | Reference | | |
| Yes | 178 | 2.417 | (1.334 4.378) | 0.004 | 2.582 | (1.409 4.731) | 0.002 |
| **Mother´s gravidity** | | | | | | | |
| 1 | 570 | Reference | | | Reference | | |
| 2 | 365 | 0.328 | (0.153 0.701) | 0.004 | 0.312 | (0.141 0.690) | 0.004 |
| 3 | 203 | 0.797 | (0.376 1.689) | 0.553 | 0.860 | (0.401 1.846) | 0.699 |
| 4 | 118 | 0.602 | (0.239 1.518) | 0.282 | 0.520 | (0.198 1.366) | 0.185 |
| 5+ | 170 | 1.250 | (0.605 2.582) | 0.547 | 1.279 | (0.609 2.686) | 0.515 |

[a]Totals across exposure categories may not equal 1494 because of missing information.

[b]IRR = Incidence rate ratio.

[c]CI = confidence interval.

[d]*Bolsa Família* = Federal conditional cash transfer program.

[e]API = annual parasite incidence in the locality of residence (continuous variable).

maternal plasma samples collected at delivery, respectively. Age-related differences in maternal antibody prevalence were significant for PvAMA1 (Mantel-Haenzel $\chi^2$ test for linear trend, $P = 0.015$), but not for other antigens (Fig 5). Levels of anti-PvAMA1 IgG antibodies were weakly, but significantly correlated with maternal age ($r_s = 0.085$, $P = 0.005$, Spearman correlation test). In contrast, no correlation with age was found for levels of anti-PvDBP IgG ($r_s = 0.024$, $P = 0.434$) or anti-PvMSP1$_{19}$ IgG ($r_s = 0.014$, $P = 0.640$).

Elevated reactivity indices of specific antibodies at delivery were not associated with protection from vivax malaria among mothers (Fig 6A). On the contrary, we noted an increased risk of vivax malaria in mothers with high antibody levels (quintile 5). Specific antibody levels were significantly and positively associated with the HR for the time to the first vivax malaria in mothers up to 12 months following delivery, after adjusting for potential confounders (Table 2). When we analyzed the period between 12 and 24 months after delivery, we found no association between specific IgG levels measured at delivery and malaria risk (Table 2).

Because vivax malaria was infrequent among study children before the age of 1 year, with only 14 children with complete serological data suffering one or more infections, the association between maternal antibodies and early life malaria could not be properly assessed (Fig 6B). Cox regression models showed no significant association between maternal antibody levels and the overall risk of malaria in children during the first two years of life (Fig 6B and

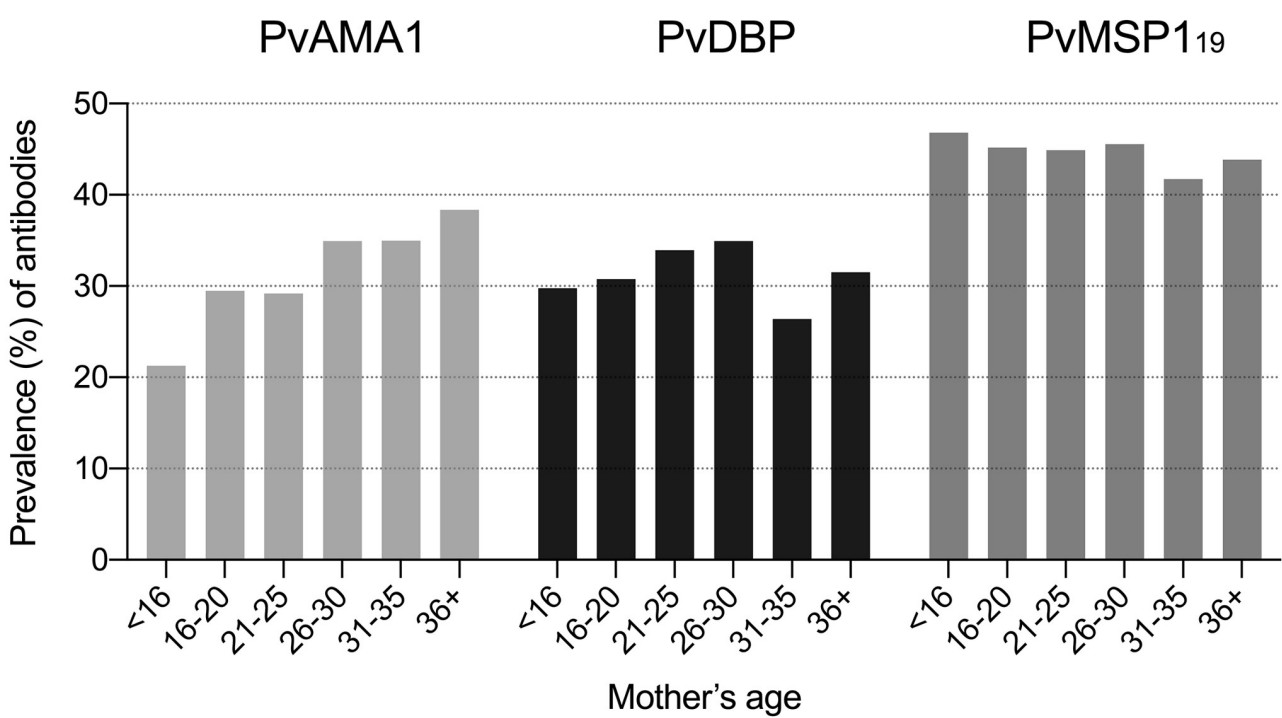

**Fig 5. Age-related prevalence of maternal IgG antibodies to the blood-stage *P. vivax* antigens PvAMA1, PvDBP, and PvMSP1_{19} in MINA-Brazil study participants.** Numbers of tested samples per age group were: <16 years, n = 47; 16–20 years, n = 312; 21–25 years, n = 274; 26–30 years, n = 226; 31–35 years, n = 163; and 36+ years, n = 73. Age-related differences in antibody prevalence were significant only for PvAMA-1 (Mantel-Haenszel $\chi^2$ test for linear trend, $P$ = 0.015).

Table 2). However, given the small number of vivax malaria episodes and the wide 95% CIs, we do not interpret these negative findings as a definitive proof of lack of any association.

## Malaria and anemia in early childhood

A total of 108 (12.6%; 95% CI, 10.5–15.0%) of the 860 children tested at approximately 2 years of age had anemia (hemoglobin concentration <110 g/L). No study participant had <70 g/L of hemoglobin, a level that defines severe anemia (https://www.who.int/vmnis/indicators/haemoglobin.pdf). Significant correlates of increased risk of anemia at the 2-year follow-up visit (among the covariates listed in S3 Fig) were: (i) increasing child age, (ii) child reporting recent vomiting, (iii) mother assisted by the *Bolsa Família* conditional cash transfer program (a proxy of, but not necessarily synonymous with poverty), (iv) mother reporting gestational hypertension, and (v) maternal anemia at delivery (Table 3). Gestational night blindness was significantly associated with decreased anemia risk, but we have no clear-cut explanation for this finding. The association between malaria since birth (regardless of the species, number, or timing of episodes) and anemia did not reach statistical significance (Table 3). However, repeated and recent malaria episodes (<12 months before the follow-up assessment), even those caused by the putatively more benign parasite *P. vivax*, significantly increased the risk of anemia at the age of 2 years (Fig 7). We note that many of the study children who are highly exposed to malaria live in hard-to-reach rural localities (Fig 4) and were not assessed for hemoglobin levels at the 2-year follow-up visit.

Quantile regression analysis further confirmed the association between repeated vivax malaria episodes and decreased hemoglobin. Results were significant for the 25th percentile

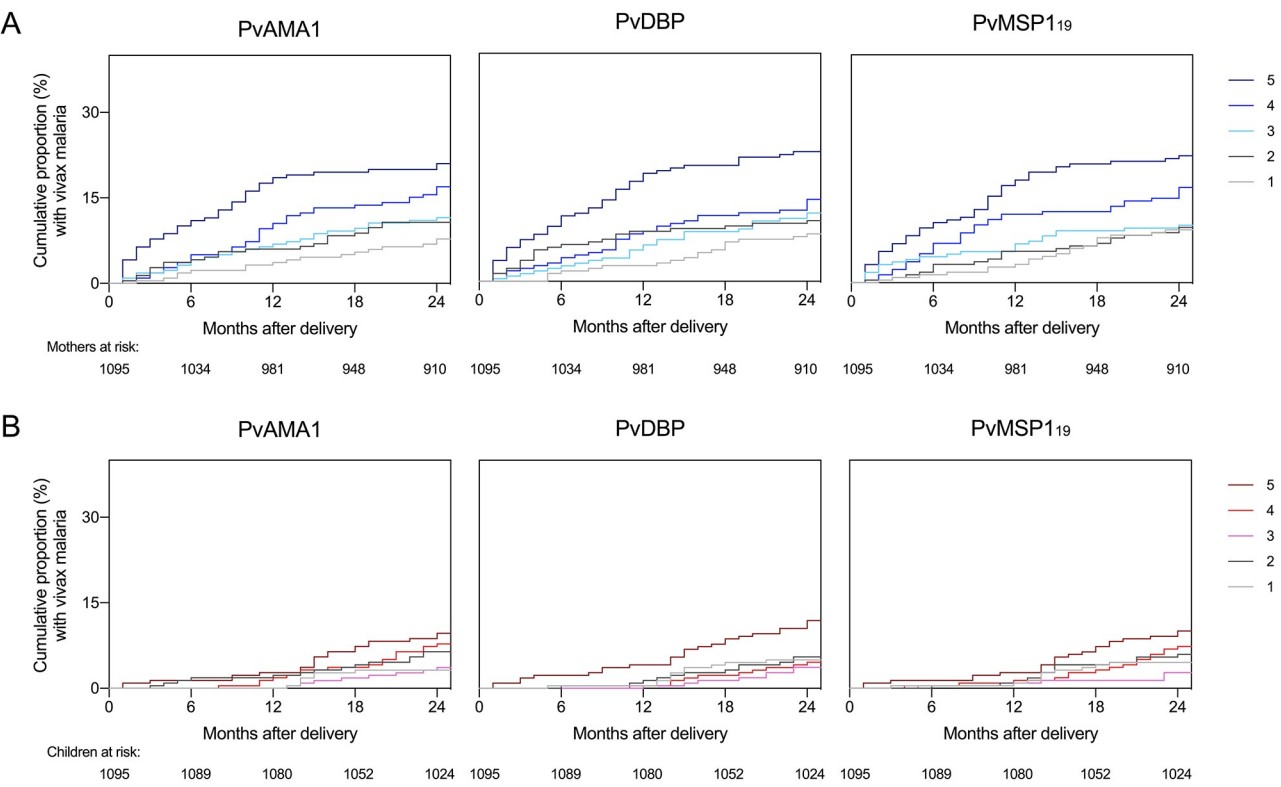

**Fig 6. Maternal antibodies and risk of *Plasmodium vivax* malaria.** Cumulative proportions of mothers (A) and their children (B) participating in the MINA-Brazil study who had vivax malaria over 2 years of follow-up are shown according to levels of antibodies to the blood-stage *P. vivax* antigens PvAMA-1, PvDBP, and PvMSP1₁₉ measured at delivery and stratified in quintiles (1 = lowest antibody levels). Children and mothers were censored at the time they had falciparum malaria.

(-5.10 g/L [95% CI, -9.71 to -0.50 g/L]) and the $50^{th}$ percentile of hemoglobin values (-6.35 g/L [95% CI, -12.44 to -0.26 g/L]), after adjusting for child age and sex, wealth index, whether the mother is beneficiary of the *Bolsa Família* program (no vs. yes), maternal schooling, whether the mother is the household head, household size, maternal anemia at delivery, and gestational age (S2 Table).

## Discussion

We have examined the burden of *P. vivax* infection in the course of the first 1000 days–from conception to child's second birthday–in the MINA-Brazil cohort study. Vivax malaria in pregnancy in this population-based cohort was previously shown to be associated with increased risk of maternal anemia and impaired fetal growth [21]. Here, we show that young infants are little exposed to, or largely protected from infection, but children >12 months of age appear to be as vulnerable to vivax malaria as their mothers. High levels of anti-*P. vivax* maternal antibodies measured at delivery do not protect mothers and their offspring from future infections. Importantly, multiple vivax malaria episodes among children are associated with increased risk of anemia in early life in this relatively low-endemicity setting.

Only 7.1% of the cohort children ever suffered a malaria episode during their first 2 years of life in the main transmission hotspot of Amazonian Brazil, and the vast majority of infections were due to *P. vivax*. However, infection risk is very heterogeneous. A handful of high-risk children had ≥4 infections and together accounted for >40% of malaria episodes diagnosed in

**Table 2. Cox proportional hazards models for time to the first vivax malaria in mothers (0–12 months after delivery [n = 1095] and 12–24 months after delivery [n = 981]) and children (24 months of follow-up since birth; n = 1095) from the MINA-Brazil birth cohort study.**

| Population | Outcome | Antigen | HR[c] according to antibody quintile | | 95% CI[d] | P |
|---|---|---|---|---|---|---|
| Mothers (n = 1095) | Time to the first vivax malaria within 1 year after delivery (96 failures)[a] | PvAMA1 | 1 | Reference | | |
| | | | 2 | 1.716 | (0.683 4.308) | 0.251 |
| | | | 3 | 1.918 | (0.773 4.755) | 0.160 |
| | | | 4 | 2.362 | (0.999 5.582) | 0.050 |
| | | | 5 | 2.935 | (1.255 6.865) | 0.013 |
| | | PvDBP | 1 | Reference | | |
| | | | 2 | 3.271 | (1.304 8.207) | 0.012 |
| | | | 3 | 1.794 | (0.671 4.796) | 0.244 |
| | | | 4 | 2.695 | (1.067 6.803) | 0.036 |
| | | | 5 | 3.755 | (1.544 9.131) | 0.004 |
| | | PvMSP1$_{19}$ | 1 | Reference | | |
| | | | 2 | 2.022 | (0.754 5.421) | 0.162 |
| | | | 3 | 2.236 | (0.848 5.898) | 0.104 |
| | | | 4 | 3.433 | (1.401 8.411) | 0.007 |
| | | | 5 | 3.672 | (1.508 8.937) | 0.004 |
| Mothers (n = 981)[e] | Time to the first vivax malaria between 12 and 24 months after delivery (51 failures)[a] | PvAMA1 | 1 | Reference | | |
| | | | 2 | 0.987 | (0.410 2.380) | 0.977 |
| | | | 3 | 1.131 | (0.480 2.665) | 0.778 |
| | | | 4 | 1.478 | (0.657 3.322) | 0.345 |
| | | | 5 | 0.534 | (0.175 1.627) | 0.269 |
| | | PvDBP | 1 | Reference | | |
| | | | 2 | 0.339 | (0.109 1.053) | 0.061 |
| | | | 3 | 1.175 | (0.542 2.545) | 0.683 |
| | | | 4 | 1.094 | (0.497 2.408) | 0.823 |
| | | | 5 | 0.711 | (0.280 1.803) | 0.472 |
| | | PvMSP1$_{19}$ | 1 | Reference | | |
| | | | 2 | 0.591 | (0.254 1.377) | 0.223 |
| | | | 3 | 0.631 | (0.272 1.460) | 0.281 |
| | | | 4 | 0.699 | (0.308 1.586) | 0.392 |
| | | | 5 | 0.659 | (0.274 1.582) | 0.350 |
| Children (n = 1095) | Time to the first vivax malaria in the first 2 years of life (67 failures)[b] | PvAMA1 | 1 | Reference | | |
| | | | 2 | 1.556 | (0.621 3.900) | 0.345 |
| | | | 3 | 1.054 | (0.381 2.915) | 0.919 |
| | | | 4 | 1.472 | (0.592 3.661) | 0.405 |
| | | | 5 | 1.092 | (0.430 2.776) | 0.853 |
| | | PvDBP | 1 | Reference | | |
| | | | 2 | 1.083 | (0.474 2.470) | 0.851 |
| | | | 3 | 0.507 | (0.200 1.282) | 0.151 |
| | | | 4 | 0.713 | (0.299 1.698) | 0.445 |
| | | | 5 | 0.975 | (0.452 2.103) | 0.948 |
| | | PvMSP1$_{19}$ | 1 | Reference | | |
| | | | 2 | 1.088 | (0.471 2.515) | 0.843 |
| | | | 3 | 0.460 | (0.165 1.281) | 0.137 |
| | | | 4 | 1.015 | (0.449 2.290) | 0.972 |
| | | | 5 | 0.814 | (0.360 1.841) | 0.621 |

[a]Cox proportional hazards model adjusted for household wealth index (quartiles), annual parasite Incidence (API) in the area of residence, and malaria in pregnancy (no vs. yes).

[b]Cox proportional hazards model adjusted for household wealth index (quartiles), annual parasite Incidence (API) in the area of residence (continuous variable), and malaria in pregnancy (no vs. yes), gravidity (0, 1, 2, 3, or 4+), and prenatal alcohol use (no vs. yes).

[c]HR = hazard ratio

[d]CI = confidence interval.

[e]Mothers who had malaria up to 12 months after delivery were excluded.

**Table 3. Multiple logistic regression analysis of correlates of anemia (hemoglobin concentration <110 g/L) at two years of age in children from the MINA-Brazil birth cohort study (n = 860).**

| Variables | n[b] | OR[c] | 95% CI[d] | P |
|---|---|---|---|---|
| | | | Outcome: anemia at the age of 2 years (n = 108) | |
| **Child´s sex** | | | | |
| Female | 429 | Reference | | |
| Male | 431 | 1.415 | (0.930 2.151) | 0.105 |
| **Child´s age (days)** | 860 | 1.005 | (1.001 1.009) | 0.024 |
| **Mother beneficiary of the Bolsa Família program[a]** | | | | |
| No | 452 | Reference | | |
| Yes | 408 | 1.860 | (1.217 2.843) | 0.004 |
| **Maternal anemia at delivery** | | | | |
| No | 494 | Reference | | |
| Yes | 320 | 1.736 | (1.133 2.659) | 0.011 |
| **Gestational night blindness** | | | | |
| No | 769 | Reference | | |
| Yes | 91 | 0.410 | (0.172 0.976) | 0.044 |
| **Child vomiting in the last 15 days** | | | | |
| No | 739 | Reference | | |
| Yes | 121 | 1.902 | (1.126 3.215) | 0.016 |
| **Malaria since birth, any species** | | | | |
| No | 828 | Reference | | |
| Yes | 32 | 2.241 | (0.965 5.205) | 0.061 |

[a]Bolsa Família = Federal conditional cash transfer program.

[b]Totals across exposure categories do not equal 860 because of missing information

[c]OR = Odds ratio.

[d]CI = confidence interval.

study participants. The highest malaria incidence was observed in rural localities (Fig 4), where effective control measures are harder to implement compared with urban and periurban settings.

Neonates and young infants <6 months of age typically experience fewer malaria episodes than older infants and children [7], although early life infections may not be uncommon in some African settings [37]. Here, we show that vivax malaria remains infrequent far beyond the age of 6 months; in fact, until the infant′s first birthday (Fig 3). Importantly, maternal antibodies transferred to the fetus and erythrocytes with high fetal hemoglobin concentration are unlikely to circulate for >6 months and account for the reduced risk in older infants [5]. Breastfeeding may confer only short-term protection to MINA-Brazil cohort participants because their median duration of exclusive breastfeeding is as short as 16 days [38].

Maternal antibodies have been hypothesized but not conclusively shown to reduce the risk of *P. vivax* infection in young infants [5]. One may argue that passively transferred immunity is unlikely to work in the infant while the mother continues to suffer from malaria, as we have observed in our cohort participants (Fig 3). However, conventional serology detects both protective and nonprotective antibodies and high antibody levels may indicate increased exposure rather than reduced risk [39]. The fine specificity and functional properties of parasite-inhibitory anti-*P. vivax* antibody responses remain largely unexplored [39,40]. The positive association between antibodies measured at delivery and subsequent malaria risk in mothers disappeared over the second year of follow-up (Table 2), suggesting that antibodies essentially

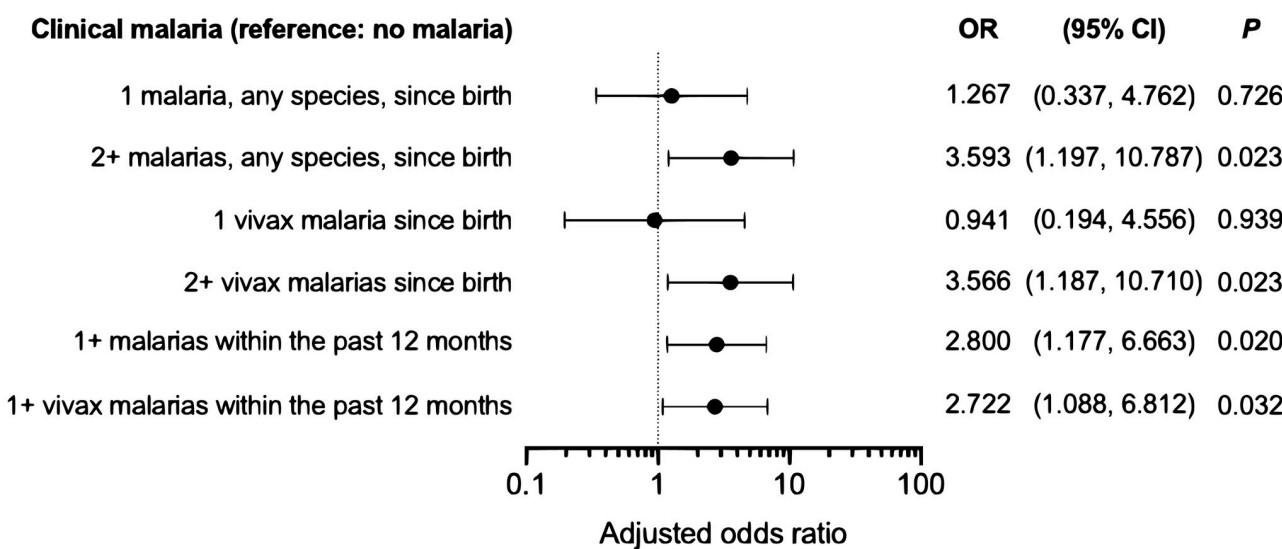

**Fig 7. Impact of the frequency and timing of malaria episodes, as well as the infecting malaria parasite species, on the risk of anemia.** Odds ratios (OR) indicate the magnitude of association between anemia at the age of 2 years and malaria type (any species or vivax), frequency (1 or 2+), or timing (since birth or within the last 12 months) in MINA-Brazil study participants (n = 860), compared with no malaria, while controlling for the potential confounders listed in Table 3. OR estimates and their respective 95% confidence intervals (95% CIs) and *P* value were derived from separate multiple logistic regression models in which exposure to malaria was introduced in different ways.

indicate recent malaria exposure, but not future long-term malaria exposure or risk. Antibody concentration may also matter for infant protection. Levels of parasite-inhibitory antibodies that fail to protect heavily exposed mothers may suffice to protect their less exposed offspring if placental transfer to the fetus during pregnancy is efficient.

We hypothesize that infants <12 months are infrequently infected because they remain largely unexposed to vectors. First, they are little affected by outdoor malaria transmission, which predominates over indoor transmission in most of the Amazon [15], until they start to walk and actively explore the external environment. Second, infants may be less attractive to mosquito vectors than older children [41], partially due to their smaller body size. Third, long-lasting insecticide-treated bed nets, which are available in high-transmission localities across the study site [42], are likely to avert more infectious bites among infants who typically sleep >12 hours per day, compared with older children.

Children born to mothers who suffered from malaria in pregnancy are at a significantly elevated risk of malaria [43]. Indeed, malaria disproportionally affects pregnant women living in transmission hotspots, where their offspring is also more likely to be infected. However, we show that this positive association persists after controlling for local API, a proxy of malaria transmission intensity in the area of residence of mother-child pairs (Table 1). We acknowledge some risk of residual confounding, since all mother-child pairs living in the same locality are assigned the same API but may still differ in malaria risk. The increased risk is not due to congenital transmission either. Indeed, congenital malaria is rare in the study site; only 6 of 637 (0.9%) neonates in this cohort tested positive for parasite DNA in cord blood samples collected at delivery [21]. Placental pathology leading to prenatal exposure to *P. falciparum* antigens may offer an explanation, as these antigens can modulate fetal immune responses [44] and induce an immune tolerant phenotype associated with increased risk of malaria and other early childhood infections [44,45]. It is debatable whether this holds for *P. vivax*, the locally dominant parasite that does not sequester massively in the intervillous spaces and causes less severe histological changes in the placenta [46].

Childhood anemia remains common in the Amazon, with 20–48% of the infants and toddlers found to be anemic and 62–81% found to be iron deficient in recent surveys in Amazonian Brazil [47,48]. Vitamin A, vitamin B12, and folate deficiencies are substantially less prevalent (4–13%) in Amazonian children [49]. Although we found much less anemia among MINA-Brazil cohort participants at the age of 2 years (prevalence, 12.6%), compared with proximate sites in the Amazon [47,48], we note that hemoglobin measurements were made in 76.6% of the study participants at the 1-year follow-up visit [16] and those with anemia (41.0% [95% CI, 37.3 to 44.8%]) were prescribed iron supplements. Treatment was unsupervised, but most anemic children may have had a sustained hemoglobin response following oral iron administration.

Importantly, anemia is widely recognized as the main adverse health outcome of vivax malaria in children [7]. Severe anemia is more prevalent in regions where malaria transmission is intense and hemoglobin concentrations are already low prior to infection, due to multiple micronutrient deficiencies [50]. Accordingly, vivax malaria accounts for nearly 28% of all episodes of moderate or severe anemia diagnosed among infants from Papua, Indonesia [13]. Despite the lower average parasite densities in *P. vivax*, compared with P. *falciparum* infections, uninfected red blood cell removal appears to be proportionally greater in vivax malaria, resulting in a similar red blood cell loss for both species [50]. The burden of anemia attributable to repeated vivax malaria largely exceeds that caused by *P. falciparum* in young children from high-endemicity areas in the Asia-Pacific region [12].

We show that repeated and recent *P. vivax* infections are associated with a significantly increased risk of anemia at the age of 2 years in a relatively low transmission setting (Fig 7). This association is likely to be causal, but residual confounding must also be considered given that both malaria and anemia tend to affect the poorest individuals in the community. Vivax malaria in pregnancy is also positively associated with maternal anemia at delivery and poorer birth outcomes in the MINA-Brazil cohort [21] and children born to anemic mothers are more likely to be anemic by their second birthday, after controlling for potential confounders (Table 3). Recurrent infections with *P. vivax* are common due to its ability to stay dormant in the liver and cause relapses within months after a single mosquito inoculation [51]. Relapses may occur shortly after the primary infection, before the child′s hemoglobin concentration has returned to normal. Although primaquine is routinely prescribed with chloroquine to suppress *P. vivax* relapses in Latin America [1], infants <6 months of age with unknown glucose-6-phosphate dehydrogenase (G6PD) deficiency status are considered primaquine-ineligible in Brazil because of the risk of hemolysis [25]. As G6PD testing is not widely available, young infants are very rarely prescribed primaquine. Moreover, not all eligible children and adults adhere to the unsupervised 7-day primaquine treatment [52] and shortened anti-relapse regimens are urgently needed [53].

The present study has some limitations. First, malaria episodes in study children were retrieved retrospectively from a case notification database and no blood samples were available for further confirmatory diagnostic tests. We assume that nearly all clinical malaria episodes confirmed by microscopy and treated in cohort participants were retrieved [23], but routine surveillance overlooks transient or chronic submicroscopic (often asymptomatic) parasitemias that do not develop into detectable infections but may cause anemia [54]. Our study design does not allow for the estimation of the adverse health effects of submicroscopic or asymptomatic infections. Second, analyses of passively detected malaria episodes are prone to biases due to differences in access to health facilities and health-seeking behavior. Third, the impact of malaria on anemia may have been underestimated because cohort participants living in remote rural areas, who are more heavily exposed to malaria and putatively more prone to its more severe consequences, were not eligible for the follow-up visit at the

age of 2 years. Fourth, because of the rarity of the outcome of interest in young infants living in urban and proximate rural areas, our study was relatively underpowered to address the association between antibody levels measured in mothers at delivery and the risk of vivax malaria in their offspring during the first months of life. Finally, the infrequency of *P. falciparum* malaria in the study population precludes between-species comparisons of malaria risk factors and clinical impact.

## Conclusion

Here, we show that the adverse effects of malaria exposure *in utero* extend into early childhood: antenatal infection in mothers is associated with an elevated risk of *P. vivax* malaria in their offspring. Repeatedly infected children are at increased risk of anemia at the age of 2 years. In conclusion, we have identified cumulative adverse effects of *P. vivax* infections that are often overlooked in low transmission settings and further challenge the traditional view of vivax malaria as a relatively benign infection in pregnancy and early childhood. There is an urgent need for improved strategies for vivax malaria prevention during the first 1000 days of life, which may include the systematic use of weekly chloroquine prophylaxis following vivax malaria treatment in pregnancy [22] and malaria screening during routine antenatal [55] and infant care in endemic settings.

## Supporting information

**S1 STROBE checklist.** STROBE Statement—Checklist of items that should be included in reports of *cohort studies*: **Low-level *Plasmodium vivax* exposure, maternal antibodies, and anemia in early childhood: population-based birth cohort study in Amazonian Brazil.**
(PDF)

**S1 Text. Supplementary Methods.**
(PDF)

**S1 Table. Multiple negative binomial regression analysis of correlates of malaria (all species) and vivax malaria from birth to two years of age in children from the MINA-Brazil birth cohort study (n = 1494).** In these models, API was entered as a categorical variable (stratified into quintiles).
(PDF)

**S2 Table. Adjusted coefficients and bootstrap 95% confidence intervals (CI) of malaria exposures associated with hemoglobin concentration (g/L) at two years of age, as estimated using quantile regression analysis for children from the MINA-Brazil birth cohort study.**
(PDF)

**S1 Fig. Levels of specific antibodies (reactivity indices) in 101 paired maternal and cordblood plasma samples from MINA-Brazil study participants.** The continuous diagonal line represents identical antibody levels in maternal and cord-blood samples. Spearman correlation coefficients were 0.814 for anti-PvAMA1 antibodies, 0.801 for anti-PvDBP antibodies, and 0.858 anti-PvMSP1$_{19}$ antibodies (*P*<0.0001 for all).
(PDF)

**S2 Fig. Hierarchical conceptual framework for selection of correlates of malaria risk in early childhood.**
(PDF)

**S3 Fig. Hierarchical conceptual framework for selection of correlates of anemia risk in early childhood.**
(PDF)

**S4 Fig. Frequency distribution of the number of malaria episodes in the study population.**
Bars show the number of laboratory-diagnosed malaria episodes per child over the first two years of life (empirical data) and the continuous line shows the negative binomial function fitted to data. Both panels show the same data, but y-axis values in the right panel are square root-transformed to improve data visualization.
(PDF)

## Acknowledgments

The authors thank all participants in the MINA-Brazil Study. The teams at the Maternity Hospital, Municipal Health Secretariat, and primary health care units of Cruzeiro do Sul are gratefully acknowledged. We thank Paola B. Marchesini (Ministry of Health, Brasília, Brazil) for granting access to the electronic malaria notification database and Natália S. Ferreira for providing non-endemic control plasmas. Investigators of MINA-Brazil Study Group comprise: Alicia Matijasevich, Bárbara H. Lourenço, Jenny Abanto, Maíra B. Malta, Marcelo U. Ferreira, Marly A. Cardoso, Paulo Augusto R. Neves, and Suely G. A. Gimeno (University of São Paulo, São Paulo, Brazil); Ana Alice Damasceno, Bruno P. da Silva, and Rodrigo M. de Souza (Federal University of Acre, Cruzeiro do Sul, Brazil); Simone Ladeia-Andrade (Oswaldo Cruz Institute, Fiocruz, Rio de Janeiro, Brazil); and Marcia C. Castro (Harvard T. H. Chan School of Public Health, Boston, USA).

## Author Contributions

**Conceptualization:** Anaclara Pincelli, Marly A. Cardoso, Marcia C. Castro, Marcelo U. Ferreira.

**Data curation:** Marly A. Cardoso, Maíra B. Malta.

**Formal analysis:** Anaclara Pincelli, Marly A. Cardoso, Igor C. Johansen, Rodrigo M. Corder, Marcelo U. Ferreira.

**Funding acquisition:** Anaclara Pincelli, Marly A. Cardoso, Marcia C. Castro, Marcelo U. Ferreira.

**Investigation:** Anaclara Pincelli, Marly A. Cardoso, Maíra B. Malta, Vanessa C. Nicolete, Marcelo U. Ferreira.

**Methodology:** Marly A. Cardoso, Vanessa C. Nicolete, Marcelo U. Ferreira.

**Project administration:** Marly A. Cardoso.

**Software:** Rodrigo M. Corder.

**Supervision:** Marly A. Cardoso, Irene S. Soares, Marcia C. Castro, Marcelo U. Ferreira.

**Visualization:** Anaclara Pincelli, Igor C. Johansen.

**Writing – original draft:** Anaclara Pincelli, Marly A. Cardoso, Irene S. Soares, Marcia C. Castro, Marcelo U. Ferreira.

**Writing – review & editing:** Anaclara Pincelli, Marly A. Cardoso, Maíra B. Malta, Igor C. Johansen, Rodrigo M. Corder, Vanessa C. Nicolete, Irene S. Soares, Marcia C. Castro, Marcelo U. Ferreira.

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
