## [Decision Letter · Decision Letter 0]

15 Apr 2021

Dear Dr. Ferreira,

Thank you very much for submitting your manuscript "Low-level Plasmodium vivax exposure, maternal antibodies, and anemia in early childhood: population-based birth cohort study in Amazonian Brazil" for consideration at PLOS Neglected Tropical Diseases. As with all papers reviewed by the journal, your manuscript was reviewed by members of the editorial board and by several independent reviewers. In light of the reviews (below this email), we would like to invite the resubmission of a significantly-revised version that takes into account the reviewers' comments. 

We cannot make any decision about publication until we have seen the revised manuscript and your response to the reviewers' comments. Your revised manuscript is also likely to be sent to reviewers for further evaluation.

Sincerely,

Kamala Thriemer

Associate Editor

Lisa Ranford-Cartwright

Deputy Editor

Reviewer's Responses to Questions

**Key Review Criteria Required for Acceptance?**

**Methods**

-Are the objectives of the study clearly articulated with a clear testable hypothesis stated?

-Is the study design appropriate to address the stated objectives?

-Is the population clearly described and appropriate for the hypothesis being tested?

-Is the sample size sufficient to ensure adequate power to address the hypothesis being tested?

-Were correct statistical analysis used to support conclusions?

-Are there concerns about ethical or regulatory requirements being met?

Reviewer #1: (No Response)

Reviewer #2: I have listed by comments pertaining to the methods in the Editorial and Data Presentation Modifications and Summary and General Comments sections of this review. Overall the methods are fairly well described, but more attention should be paid to their limitations as I have outlined.

Reviewer #3: (No Response)

**Results**

-Does the analysis presented match the analysis plan?

-Are the results clearly and completely presented?

-Are the figures (Tables, Images) of sufficient quality for clarity?

Reviewer #1: (No Response)

Reviewer #2: I have listed by comments pertaining to the results in the Editorial and Data Presentation Modifications and Summary and General Comments sections of this review.

Reviewer #3: (No Response)

**Conclusions**

-Are the conclusions supported by the data presented?

-Are the limitations of analysis clearly described?

-Do the authors discuss how these data can be helpful to advance our understanding of the topic under study?

-Is public health relevance addressed?

Reviewer #1: (No Response)

Reviewer #2: I have listed by comments pertaining to the conclusions in the Editorial and Data Presentation Modifications and Summary and General Comments sections of this review. I have highlighted areas where I believe the limiations of the study need to be emphasised more.

Reviewer #3: (No Response)

**Editorial and Data Presentation Modifications?**

Reviewer #1: (No Response)

Reviewer #2: L45: Specify the actual number rather than just >1,500. 

L94: Please check the 800,000 number in the WHO report, a quick look suggests it should be 889,000 (822,000-970,000) but I may be wrong. 

L100-103: Worth mentioning here that reduced exposure to vectors is also a proposed mechanism. 

L119-125: These are results, suggest highlighting the primary research question(s) of the study here instead.

L165-166: The reference does not support this assertion and highlights that the poorest families were most likely to be lost to followup: “Compared with participants at birth, the proportion (95% CI) of participants from poorest families declined from 24.9% (22.5 to 27.4) at birth to 19.3% (16.8 to 22.1) at 2 years”. This should be acknowledged. 

L245-246: Quartiles are the 25th, 50th and 75th percentiles. If you cut the data at the quartiles you stratify it into fourths. 

L245-263: What was the assumed functional form of the relationship between continuous covariates and the outcome? Linear? 

L267-268: Can the authors further explain how a missing value category is incorporated into a continuous covariate? 

L276 – quintiles are the 20th, 40th, 60th and 80th percentiles. I think the authors are referring to “fifths”. Can the authors also justify their choice of categorising the antibody exposure into fifths rather than retaining the continuous information? 

L286 – typo: “area or residence”

L328 – Can the authors clarify if an infection of both P. vivax and P. falciparum considered as a P. vivax infection in the data analysis? 

Figure 5A: Why not display antibody RI on the y axis and continuous age on the x axis? 

Figure 5B: Can the authors explain why they have plotted their reactivity index data on a linear scale? Given that it is a ratio I would expect to see it on a log scale. 

Figure 6: The image upload was of poor quality. It was difficult to discern the colours but his may be an upload issue. 

Table 2 – Please clarify which fifth is the highest level and what is the lowest level. 

L389-391: “associated with decreased malaria risk” – is this a typo? I believe the outcome in this section is anemia. 

L389: Can the authors explain why maternal night blindness was considered as a risk factor for inclusion in the model? 

Table 3 – Please five the units for age. 

L444-451: An alternative explanation is that the API used in the model reflects an average API across an area that will not apply equally to all women and the women from an area with a given API who have experienced an infection during pregnancy are probably on average at greater risk of exposure and infection than those women from the same API area who did not experience an infection. Also worth noting that the model has assumed that the relationship between API and childhood malaria is linear.

Reviewer #3: (No Response)

**Summary and General Comments**

Reviewer #1: This useful paper presents data collected as part of a large cohort study of mother-baby pairs in Amazonian Brazil. In this case, the focus of the analysis is morbidity from, and risk factors for, Plasmodium vivax malaria in children and mothers in the two years post delivery. Follow-up was primarily passive, relying on routinely-collected diagnostic data reported to the Brazilian Ministry of Health. A subset of over 300 remote-living mother-child pairs was excluded from many analyses as these individuals could not be reached for subsequent questionnaire administration and blood sampling at the two-year mark. A large proportion of mothers and children did not provide blood specimens at delivery. Despite these shortcomings, the analyses are robust and the results of significant interest. Salient findings include the relatively small number of children (7.1%) experiencing vivax malaria before their second birthday, the apparent protection against vivax malaria out to 1 year of age (substantially longer than passively transferred maternal antibodies could be expected to be present) and the marked concentration of cases in a small number of high-risk individuals. Repeated episodes of vivax malaria was associated with anaemia, something that has oft been postulated but not proved. 

I have some fairly minor comments that I hope will be helpful for the authors.

1. Abstract, line 45: give a specific denominator here, not “>1,500 participants”

2. Abstract: the abstract essentially presents no methodology, other than to say the study was a “birth cohort”. At a minimum, the reader needs to know whether malaria detection was passive or active (passive in this case) and what means of diagnosis was used (ie microscopy, RDT, PCR). I would also suggest stating that a questionnaire was administered, and blood sampled at a two-year follow-up visit to mother and child.

3. Suggest use term “children under 5 years old” rather than “under-five children” to avoid ambiguity

4. Introduction: The salient results of the paper are presented in the last part of the introduction, before any of the methodology. This is very unorthodox and means the reader cannot appraise the validity of the results as they do not know how they were produced. I would suggest moving this result summary to the start of the discussion section.

5. Methods: In the methods section, the authors state that serology was only done on 101 cord blood specimens. In the results section it becomes apparent that 637 cord blood samples were collected. Why the discrepancy? It would also be useful to document whether all mothers of the 6 neonates with congenital malaria were parasitaemic at delivery. 

6. Methods, lines 234-236: Data on malaria episodes were collected passively and therefore cannot strictly be used to calculate prevalence. Suggest use the term “monthly incidence” (as per the excellent Figure 3).

7. Methods: statistics generally well-described but there is no mention of the validity of the proportionality assumption for the Cox models. To my eye, the Kaplan-Meier curves suggest significant non-proportionality. How was this dealt with?

8. Methods: there is no mention of antimalarial treatment protocols in the methods section – this only appears late in the discussion. In the methods I would suggest presenting the standard treatment protocol, at least for P. vivax, and providing details about eligibility criteria for primaquine. The relevant section in the discussion suggests to me that infants <6 months of age with documented normal G6PD activity are prescribed primaquine (outside of WHO guidelines but quite possibly the right thing to do). Is this correct?

9. Results and discussion: I would caution against implying causality when referring to the association between repeat vivax malaria events and subsequent anaemia. Although there is biological plausibility for this, there are many other confounders that may be relevant.

10. Results: The authors lapse into speculation in several places in the results section. I would present the facts in the results and leave the speculation to the discussion.

11. Conclusion (both abstract and main paper): I think the conclusion that repeated infections and subsequent anaemia impede normal growth and development cannot be claimed on the basis of the results of the current study. I would suggest removing the phrase “which impedes their normal growth and development”.

12. Figure 4: Perhaps an editorial problem, but figure 4 seemed to be very low resolution and I could not read the text.

Reviewer #2: Summary

Pincelli et al have conducted an interesting study on a prospective birth-cohort in Brazil. Specifically the focus of this paper is on factors associated with malaria in children over two years; malaria in mothers over two years and anemia in children at two years. A particular focus is on the exposure of P. vivax antibodies as an exposure variable. I congratulate the authors on putting together this impressive piece of work. The data presented within this paper are valuable for the community. My primary concerns are that the authors place a little too much confidence in their ability to account for confounding. Studies seeking to relate malaria-specific antibodies to malaria related outcomes will always face difficulty accounting for the confounding of overall malaria and vector exposure. As such I think the authors need to be a little bit more circumspect in their conclusions and acknowledge the possibility of residual confounding explaining some of their results. 

General comments

The authors use unadjusted bivariable analysis to screen risk factors for use in multivariable analysis and have only retained covariates which have associations with the outcome at a significance level of 10%. There needs to be some discussion of the shortcomings of this approach. I recommend the following articles: https://www.sciencedirect.com/science/article/pii/089543569600025X

https://doi.org/10.2105/AJPH.79.3.340

https://journalofbigdata.springeropen.com/articles/10.1186/s40537-018-0143-6

 “Malaria in pregnancy emerged as a significant correlate of subsequent risk of malaria in the offspring, which is not prevented by maternal anti-P. vivax antibodies transferred to the fetus.”

I think the conclusion that maternal anti-P. vivax antibodies are not preventing any detected malaria in the offspring is not warranted given the data. My two main reasons for this belief are:

1. Given the extremely strong relationship between levels of maternal antibodies and maternal malaria I am sceptical that there is no residual confounding of malaria exposure risk for the mother (and child). 

2. Even if the model had perfectly accounted for all confounders (or if these estimates were from an RCT) then I would not conclude that there is no association. The lower bounds of the 95% CI for each vivax antibody upper fifth are 0.4, 0.5 and 0.4 so it would be inappropriate to rule out a protective effect, simply put the uncertainty around the estimates is large and the data are compatible with clinically relevant associations in both direction - see https://www.bmj.com/content/311/7003/485.full

The authors could consider modelling haemoglobin levels directly as a supplementary analysis, rather than dichotomising into anaemia yes/no as this would improve power.

Reviewer #3: (No Response)

PLOS authors have the option to publish the peer review history of their article (what does this mean?). If published, this will include your full peer review and any attached files.

Reviewer #1: Yes: Nicholas M Douglas

Reviewer #2: No

Reviewer #3: No
---

## [Decision Letter · Decision Letter 1]

16 Jun 2021

Dear Dr. Ferreira,

We are pleased to inform you that your manuscript 'Low-level Plasmodium vivax exposure, maternal antibodies, and anemia in early childhood: population-based birth cohort study in Amazonian Brazil' has been provisionally accepted for publication in PLOS Neglected Tropical Diseases.

IMPORTANT: The editorial review process is now complete. PLOS will only permit corrections to spelling, formatting or significant scientific errors from this point onwards. Requests for major changes, or any which affect the scientific understanding of your work, will cause delays to the publication date of your manuscript. I note that one of the reviewers had made additional suggestions to improve the manuscript. Some of these changes could be relatively minor and so allowed, but if you wish to make any larger changes to the manuscript based on these comments then we may have to request permission from PLOS to do this.

Best regards,

Kamala Thriemer

Associate Editor

Lisa Ranford-Cartwright

Deputy Editor

Reviewer's Responses to Questions

**Key Review Criteria Required for Acceptance?**

**Methods**

-Are the objectives of the study clearly articulated with a clear testable hypothesis stated?

-Is the study design appropriate to address the stated objectives?

-Is the population clearly described and appropriate for the hypothesis being tested?

-Is the sample size sufficient to ensure adequate power to address the hypothesis being tested?

-Were correct statistical analysis used to support conclusions?

-Are there concerns about ethical or regulatory requirements being met?

Reviewer #1: (No Response)

Reviewer #2: (No Response)

Reviewer #3: (No Response)

**Results**

-Does the analysis presented match the analysis plan?

-Are the results clearly and completely presented?

-Are the figures (Tables, Images) of sufficient quality for clarity?

Reviewer #1: (No Response)

Reviewer #2: (No Response)

Reviewer #3: (No Response)

**Conclusions**

-Are the conclusions supported by the data presented?

-Are the limitations of analysis clearly described?

-Do the authors discuss how these data can be helpful to advance our understanding of the topic under study?

-Is public health relevance addressed?

Reviewer #1: (No Response)

Reviewer #2: (No Response)

Reviewer #3: (No Response)

**Editorial and Data Presentation Modifications?**

Reviewer #1: (No Response)

Reviewer #2: (No Response)

Reviewer #3: (No Response)

**Summary and General Comments**

Reviewer #1: The authors have adequately addressed my comments/suggestions in their revised manuscript. I have no further suggestions to make.

Reviewer #2: I thank the authors for their responses, and believe the paper is much improved. I have three issues that have arisen in this revision that I believe they should address:

1. Can the authors state their rationale in text for considering an individual infected with both P. vivax and P. falciparum as not having P. vivax for the purpose of this analysis? It would also be worth flagging in discussion that a factor that increases your risk of getting P. falciparum could potentially seem protective against P. vivax under this model.

2. I suggest a more nuanced interpretation of Table 2 than “Cox regression models showed no significant association between maternal antibody levels and risk of malaria during the first two years of life”. While this is a true statement it may confuse readers who frequently confuse “no significant association” with “proof of no association”. Given that the 95% CIs for this are wide (and the associations of the top fifth versus bottom fifth are not significantly different from an HR=0.5 for any anytibody) I think there should be some statement about the wide 95% CIs, they indicate considerable uncertainty and the data are compatible with strong protective associations and strong negative associations.

3. Having reviewed the Victoria et al 1997 paper and S2 Fig, I think if the following is correct then Tables 1 and 3 need some clarification to avoid reader confusion - “Estimates of incidence rate ratio are provided along with 95% CIs to quantify the influence of a given predictor on the outcome, while controlling for all other covariates in the same or more distal hierarchical level.”

In Table 1 wealth index quartile is more distal than gravidity, does this mean the estimate for wealth index quartile does not have gravidity as a covariate? This would be worth clarifying in the figure legend as most readers will expect that this is output from a single model. The same comment applies to Table 3.

Reviewer #3: The authors have adequately revised the manuscript and relevant adjunct files in response to the review comments.

PLOS authors have the option to publish the peer review history of their article (what does this mean?). If published, this will include your full peer review and any attached files.

Reviewer #1: **Yes: **Nicholas M Douglas

Reviewer #2: No

Reviewer #3: No

---

## [Editor Report · Acceptance letter]

30 Jun 2021

Dear Dr. Ferreira,

We are delighted to inform you that your manuscript, "Low-level Plasmodium vivax exposure, maternal antibodies, and anemia in early childhood: population-based birth cohort study in Amazonian Brazil," has been formally accepted for publication in PLOS Neglected Tropical Diseases.

Best regards,

Shaden Kamhawi

co-Editor-in-Chief

Paul Brindley

co-Editor-in-Chief
